# Nickel catalyzed C-N coupling of haloarenes with B₂N₄ reagents

Qianqian Chang[1,5], Qini Li[1,5], Yi-Hui Deng[2,3,5], Tian-Yu Sun [2,3] ✉,
Yun-Dong Wu [2,3,4] ✉ & Leifeng Wang [1] ✉

Carbon-heteroatom bond (especially for C-N bond) formation through nickel catalysis has seen significant development. Well-established Ni(0)/Ni(II) redox cycle and photoinduced Ni(I)/Ni(III) redox cycle have been the dominant mechanisms. We report a thermally driven Ni-catalyzed method for C-N bond formation between haloarenes and B₂N₄ reagents, yielding *N,N*-dialkylaniline derivatives in good to excellent yields with broad functional group tolerance under base-free conditions. The catalytic protocol is useful for base-sensitive structures and late-stage modifications of complex molecules. Detailed mechanistic studies and density functional theory (DFT) calculations indicate that a Ni(I)/Ni(III) redox cycle is preferred in the C-N coupling process, and B₂N₄ reagent serves both as a single electron transfer donor and a *N,N*-dialkylation source.

*N*,*N*-dialkylanilines are key scaffolds in bioactive compounds[1–3] and important building blocks for organic synthesis[4–9]. The development of more efficient and sustainable *N*-alkylation processes has continuously attracted the attention of chemists[10–13]. While direct *N*-alkylation with alkyl halide (e.g., toxic methyl iodide)[14] and reductive amination[15,16] remains in use, transition-metal (Pd/Ni etc.) promoted C-X amination reactions utilizing dialkylamines or their equivalences through well-established M(0)/M(II) redox cycle[17–22] have emerged as effective alternatives (Fig. 1a). Productive catalysis through the M(0)/M(II) cycle has been achieved through elegant ligand designs based on phosphorous[23–25] or N-heterocyclic carbenes (NHC)[26–29]. In recent years, Ni(I)/Ni(III)[30–34] redox cycle has been confirmed to be more efficient and versatile since high-valent Ni(III) species lead to a faster and energetically downhill C-N reductive elimination (RE)[28]. Recent developments in photoredox catalysis[35–39] (Fig. 1b) have demonstrated that C-N bond formation can be achieved by using simple ligands (even without ligands) under exceptionally mild reaction conditions. MacMillan[40,41] and Buchwald[42] developed a nickel/photochemical C-N bond formation methodology without complex ligands between haloarenes and amines. Miyake[43] also demonstrated a photocatalyst-free C-N cross coupling of electron deficient haloarenes via ultraviolet light photoexcitation of Nickel-amine complexes. In the absence of additional photocatalysts, both Xue[44,45] and our group[46] have reported the strategies for C-N bonds construction via photo-induced nickel catalysis. Most recently, Ritter[47] reported a Ni(I)/Ni(III)-photo-catalyzed C-heteroatom bond formation of pre-functionalized arylthianthrenium salts based on simple Ni(II) salt.

While these approaches have found wide use for C-N bond formation, they are primarily hampered by their inability to efficiently react with electron rich haloarenes and base sensitive substrates, since most of these reactions require the consumption of super-stoichiometric amounts of exogenous bases as sacrificial reagents. Still remains undeveloped, there are limited reports of base-free thermally sustained Ni(I)/Ni(III) coupling strategies, utilizing external single-electron reductants such as manganese and zinc[48–51].

While B₂O₄ type diboron reagents, such as tetrahydroxydiboron (B₂(OH)₄), are applied in the Suzuki-Miyaura coupling reaction[52,53] for C-C bond construction, bis(pinacolato)diboron (B₂pin₂) and B₂pin₂ derivatives (B₂cat₂, B₂nep₂ etc.), are often used in C-B bond formation through Miyaura borylation reaction (Fig. 1c)[54,55]. Diboron compounds

[1]School of Pharmaceutical Sciences (Shenzhen), Shenzhen Campus of Sun Yat-sen University, Shenzhen, P. R. China. [2]Key Laboratory of Computational Chemistry and Drug Design, State Key Laboratory of Chemical Oncogenomics, Shenzhen Key Laboratory of Chemical Genomics, School of Chemical Biology and Biotechnology, Peking University Shenzhen Graduate School, Shenzhen, Guangdong, P. R. China. [3]Institute of Chemical Biology, Shenzhen Bay Laboratory, Shenzhen, China. [4]College of Chemistry and Molecular Engineering, Peking University, Beijing, China. [5]These authors contributed equally: Qianqian Chang, Qini Li, Yi-Hui Deng. ✉e-mail: Tian-Yu_Sun@pku.edu.cn; wuyd@pkusz.edu.cn; wanglf33@mail.sysu.edu.cn

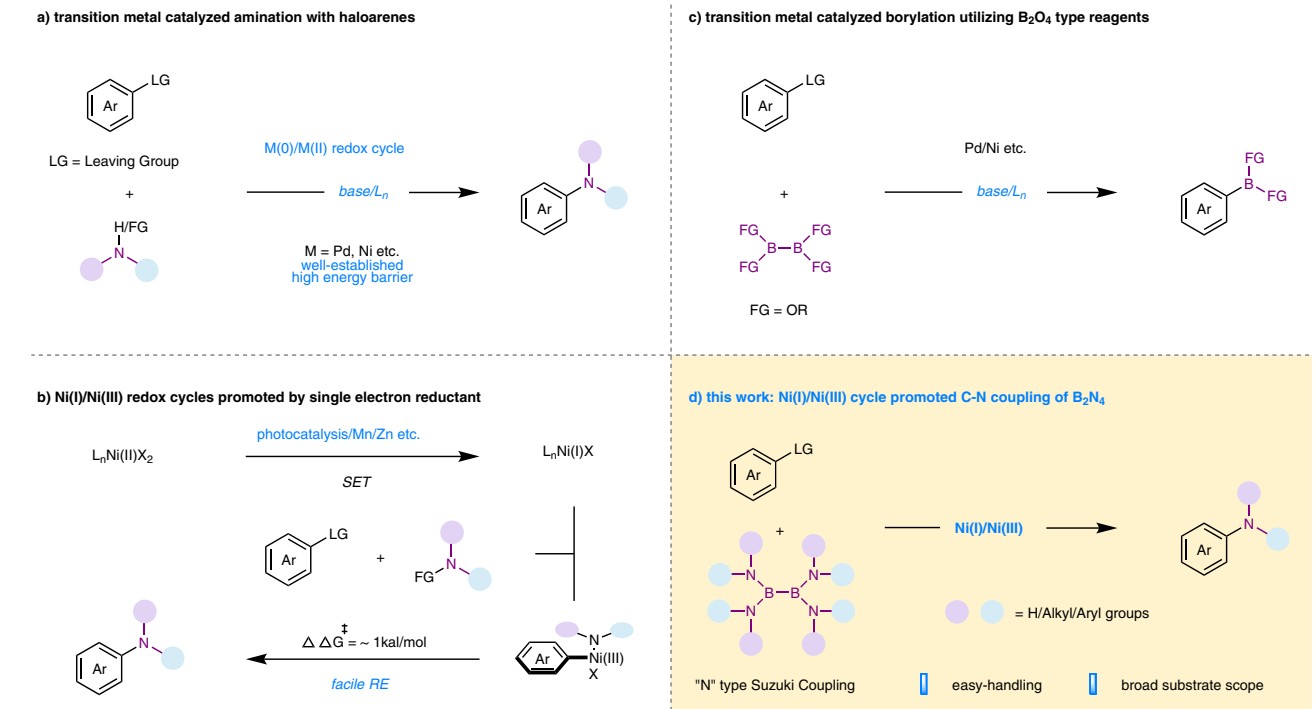

**Fig. 1 | Transition-metal catalyzed C-N bond formation of haloarenes.**
**a** transition metal catalyzed amination with haloarenes. **b** Ni(I)/Ni(III) redox cycles promoted by single electron reductant. **c** transition metal catalyzed borylation utilizing $B_2O_4$ type regents. **d** Ni(I)/Ni(III) cycle promoted C-N coupling of $B_2N_4$. LG leaving group, FG functional group.

## Table 1 | Optimization of reaction conditions

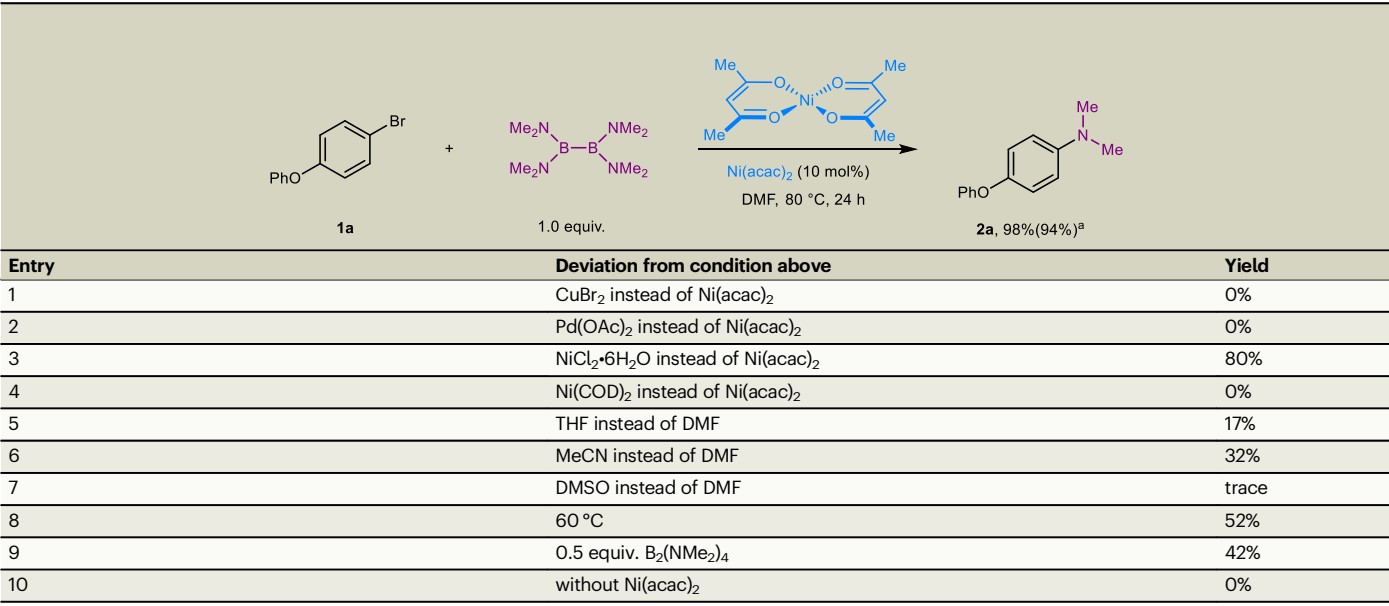

| Entry | Deviation from condition above | Yield |
|---|---|---|
| 1 | CuBr$_2$ instead of Ni(acac)$_2$ | 0% |
| 2 | Pd(OAc)$_2$ instead of Ni(acac)$_2$ | 0% |
| 3 | NiCl$_2$·6H$_2$O instead of Ni(acac)$_2$ | 80% |
| 4 | Ni(COD)$_2$ instead of Ni(acac)$_2$ | 0% |
| 5 | THF instead of DMF | 17% |
| 6 | MeCN instead of DMF | 32% |
| 7 | DMSO instead of DMF | trace |
| 8 | 60 °C | 52% |
| 9 | 0.5 equiv. B$_2$(NMe$_2$)$_4$ | 42% |
| 10 | without Ni(acac)$_2$ | 0% |

Reactions were performed with 0.2 mmol 4-bromophenoxybenzene, 0.2 mmol B$_2$(NMe$_2$)$_4$, 0.5 mL solvent and 10 mol% transition metal for 24 h under 80 °C; NMR yields using pyrazine as internal standard[a]. [a]Isolated yields.

are common and useful single electron transfer (SET) reagents[56,57] in photoredox catalysis, but they are less used as SET reagents in transition-metal catalysis. While seminal reports[58–60] in the field of C-C/C-X bonds formation utilizing $B_2O_4$ type diboron reagents have been accomplished, in sharp contrast, $B_2N_4$-type diboron reagents, such as tetrakis(dimethylamino)diboron (B$_2$(NMe$_2$)$_4$), have not been found to be useful for transition-metal-catalyzed C-N bond construction. Could a $B_2N_4$ type diboron reagent be used both as a SET reagent and a "N"

source by the Ni(I)/Ni(III) redox cycle without photoexcitation under base-free and mild reaction conditions? Herein, we report such a case, as shown in (Fig. 1d). Remarkably, nickel offers distinct advantages over palladium, such as earth-abundant metal catalyst and a reduced propensity for β-hydride (or β-hydrogen) elimination[61] when working with alkyl fragments.

*N,N*-dimethyl anilines are most valuable intermediates among *N,N*-dialkylanilines used for the preparation of bioactive molecules,

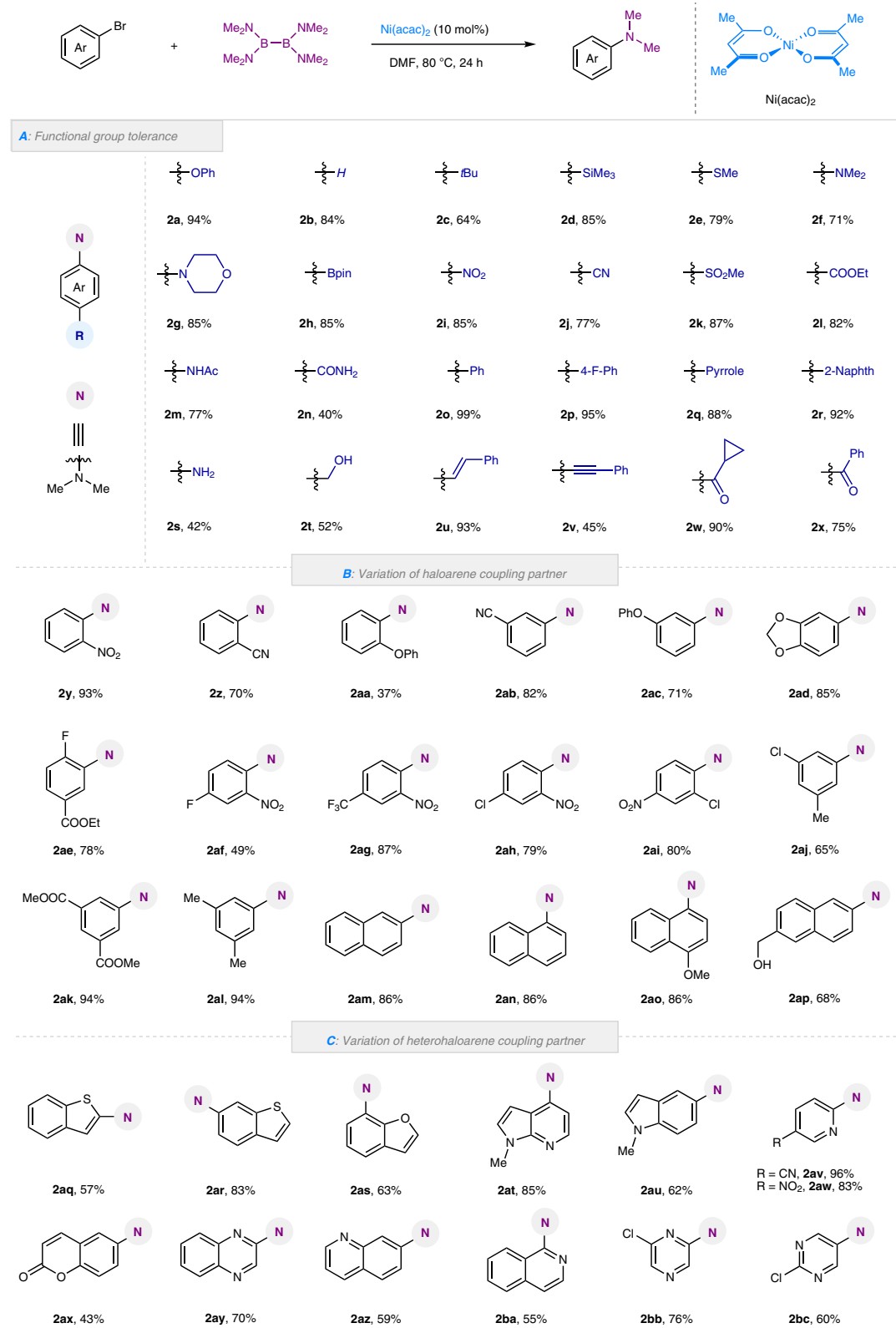

**Fig. 2 | Synthetic scope of nickel catalyzed *N,N*-dimethylation of bromoarenes.** **A** The functional group tolerance of *para*-substituted haloarenes; **B** Variation of haloarene coupling partner; **C** Variation of heterohaloarene coupling partner.

Reaction condition: substrates (0.2 mmol, 1.0 equiv.), Ni(acac)$_2$ (0.02 mmol, 10 mol %), B$_2$(NMe$_2$)$_4$ (0.2 mmol, 1.0 equiv.), DMF (0.5 mL), stirring at 80 °C for 24 h. Yields are reported for material obtained following purification and isolation.

polyester resins and dyes. Direct *N,N*-dimethylation of haloarenes, especially electron rich haloarenes, with dimethylamine is more challenging because of its low boiling point (~7 °C). Several protocols[62–65] for the dimethylamination of haloarenes using the equivalents of

dimethylamine have been reported as well. However, the pre-functionalization of dimethylamine, the requirement for strong base and complex ancillary ligands are necessary, making it nontrivial for high site selectivity. In this work, our photo-free "two-in-one" protocol

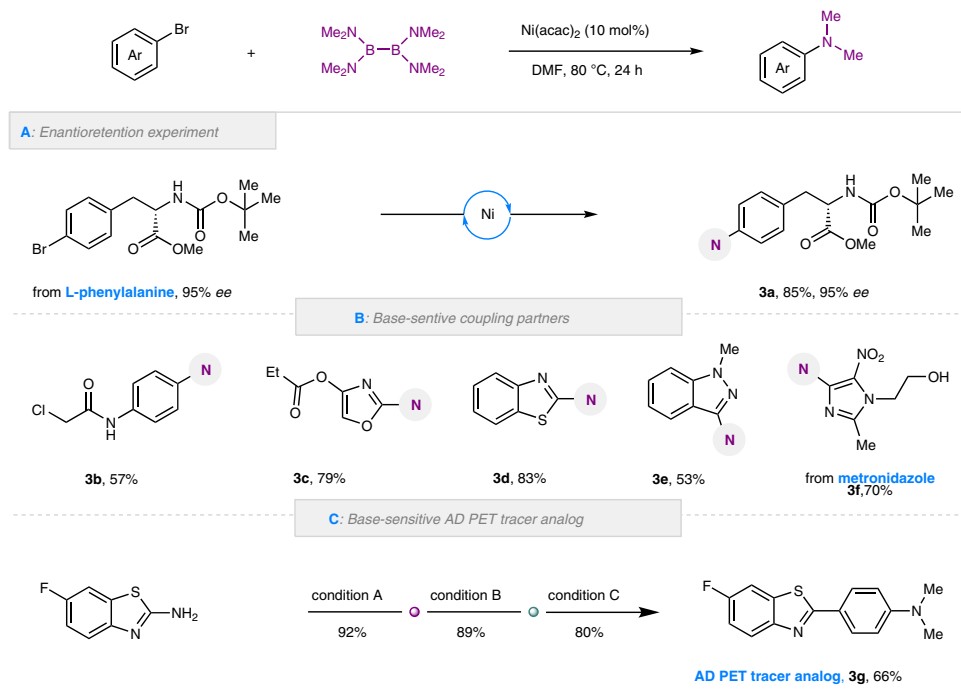

**Fig. 3 | Application of mild and base-free conditions in the functionalization of substrates prone to decomposition and racemization. A** Enantioretention experiment. **B** Base-sensitive coupling partners. **C** Synthesis of base-sensitive AD PET tracer analog. Condition A: 2-amino-6-fluorobenzothiazole (1.0 equiv.), ethylene glycol (4 equiv.), KOH (6.0 M, 30 equiv.), reflux at 140 °C for 24 h; condition B: bis(2-amino-5-fluorophenyl)disulfide (1.0 equiv.), 4-bromobenzaldehyde (1.02 equiv.), sodium metabisulfite (1.02 equiv.), DMSO (0.17 M), reaction at 120 °C for 2 h; condition C: 2-(4-Bromophenyl)−6-fluorobenzothiazole (0.2 mmol), $B_2(NMe_2)_4$ (0.2 mmol), Ni(acac)$_2$ (10 mol%), DMF (0.5 mL), reaction for 24 h under 80 °C. AD Alzheimer's disease, PET the positron emission computed tomography. See SI for full experimental details and conditions.

makes it complementary to other C-N bond formation strategies (especially for syntheses of N,N-dimethyl anilines), whose effectiveness is exemplified through working with base-sensitive partners and late-stage modifications of a series of bioactive complex molecules.

## Results and discussion

### Reaction design and optimization

Building on this basis, we set out to examine the possibility of base-free, complex ligand-free Ni-catalyzed C-N coupling of haloarenes with commercially available $B_2(NMe_2)_4$, aiming to establish a protocol that displays an effective N,N-dimethyl aniline synthesis. In our initial study, the coupling reaction of 4-bromophenoxybenzene (**1a**) with $B_2(NMe_2)_4$ was selected as a model system for reaction development, as shown in (Table 1). Various transition metal catalysts were systematically evaluated for their efficacy in our study (Table S1). These experiments demonstrated that the N,N-dimethyl aniline product **2a** formed with 94% isolated yield in the presence of 10 mol % of Ni(acac)$_2$ catalyst. Copper and palladium catalysts showed no reactivity for this reaction (entries **1-2**). The yield of **2a** was slightly reduced with NiCl$_2$•6H$_2$O as catalyst (entry **3**). In contrast, Ni(0) catalysts, for example Ni(COD)$_2$, did not show activity (entry **4**). Thus, Ni(II) is indispensable for optimizing reaction efficiency in this C-N cross coupling reaction. Subsequent studies revealed DMF to be the optimal solvent for the coupling reaction (entries **5-7**). Lower reaction temperature and reduced loading of $B_2(NMe_2)_4$ significantly diminished the yield of **2a** (entries **8-9**). Control experiments revealed that no reaction occurred without Ni(II) catalyst (entry **10**).

### Scope of substrates

With the optimal condition in hand, we first investigated the scope and limitations of our Ni-catalyzed C-N coupling reaction with respect to the para-substituted bromoarenes (Fig. 2A). This method showed excellent functional group tolerance, both electron-donating and electron-withdrawing substituents could be well accommodated, including alkyl (**2c**), alkenyl (**2u**), alkynyl (**2v**), thioether (**2e**), free hydroxyl and amino (**2s, 2t**), cyclopropyl (**2w**), (hetero)aryl (**2o-2r**). Carbonyl functionalities such as esters (**2l**), amides (**2m-2n**) and ketones (**2w-2x**) that might be susceptible to acylation or condensation reaction were not affected when $B_2(NMe_2)_4$ was used. Nitrogen-, oxygen-, sulfur-, boron-, silicon-containing haloarenes were also prepared in high yields.

We next extended the protocol to the synthesis of poly-substituted aniline derivatives, bearing in mind their importance in medicinal chemistry (Fig. 2B). Bromoarenes bearing electron-donating and electron withdrawing groups at ortho and meta position were all compatible to give the corresponding anilines in good to excellent yields (**2y-2ac**), bulky ortho-OPh-bromoarenes (**2aa**), albeit with slightly diminished yields. Di-substituted bromoarenes with electron-donating and electron-withdrawing groups, such as 3,4- 2,4- 2,5 and 3,5-disubstituted bromoarenes could give good to high yields (**2ad-2al**), showcasing the compatibility of our method with a broad array of functionality. Naphthyl bromides (**2am-2ap**) were also good partners to afford desired products. Given the critical role of heterocyclic motifs in drug development, we extended this methodology to diverse heteroaryl bromides, spanning benzothiophene, benzofuran, indole, pyridine, coumarin, quinoxaline, (iso)quinoline, pyrazine, and pyrimidine scaffolds. All of these substrates yielded the expected products in good to excellent yields (**2aq-2bc**, Fig. 2C).

Due to the mild and base-free reaction conditions, we envisioned that our strategy might be applicable to substrates that would readily undergo undesired racemization, elimination and decomposition reactions in the presence of strong bases. Chiral substituents at the α-carbonyl position are highly susceptible to epimerization, while the presence of strong bases can further induce decomposition. Armed with our standard conditions, no ee erosion was observed as

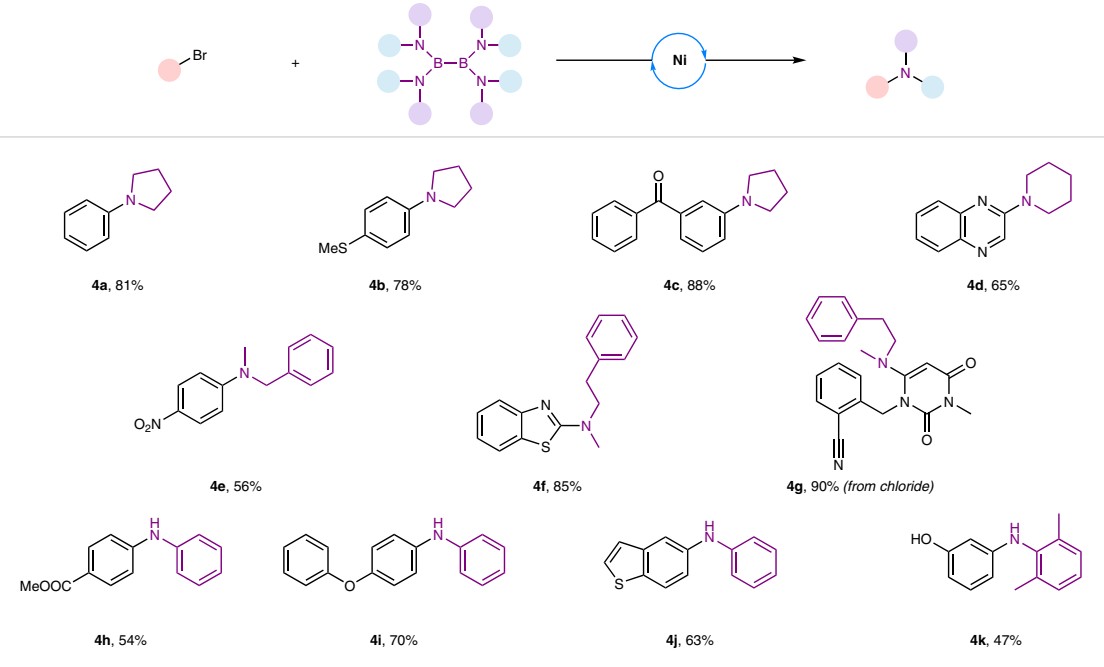

**Fig. 4 | C-N bond formations with diverse B₂N₄ reagents.** Reaction condition: substrates (0.2 mmol, 1.0 equiv.), Ni(acac)₂ (0.02 mmol, 10 mol%), B₂(NMe₂)₄ (0.2 mmol, 1.0 equiv.), DMF (0.5 mL), stirring at 80 °C for 24 h. See SI for full experimental details and conditions.

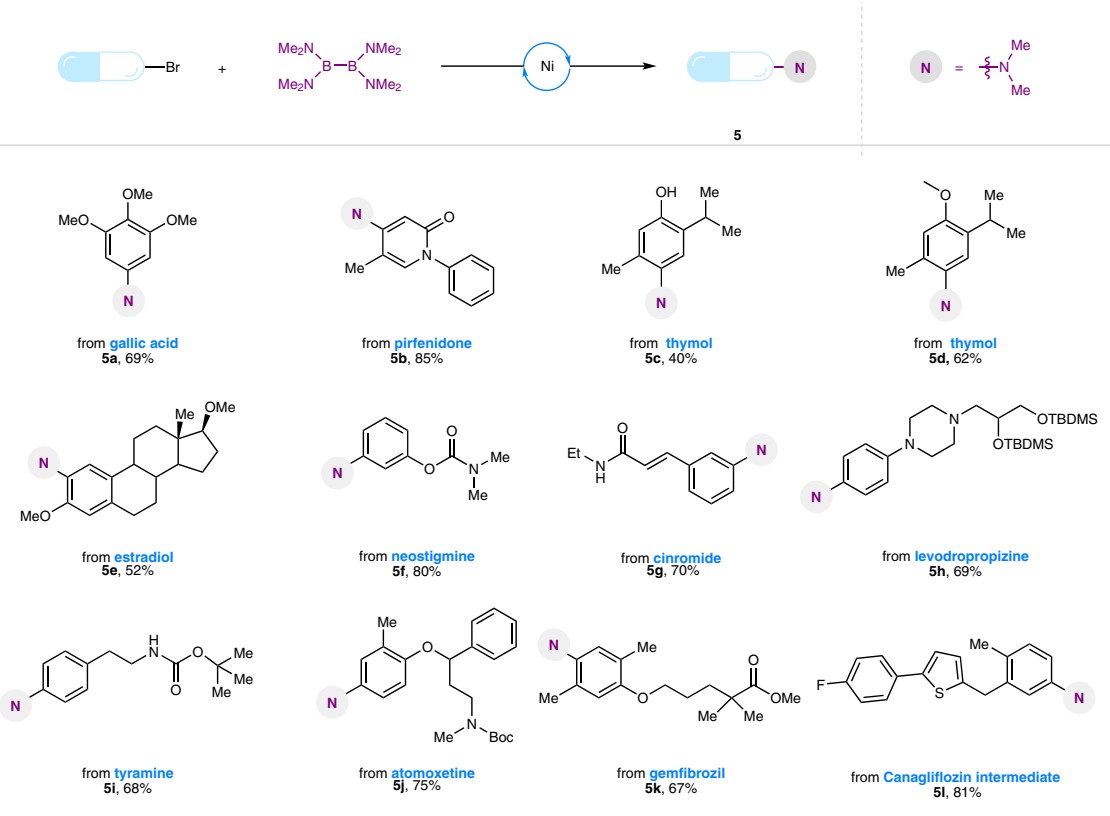

**Fig. 5 | Late-stage *N,N*-dimethylation of complex molecules.** Reaction condition: substrates (0.2 mmol, 1.0 equiv.), Ni(acac)₂ (0.02 mmol, 10 mol%), B₂(NMe₂)₄ (0.2 mmol, 1.0 equiv.), DMF (0.5 mL), stirring at 80 °C for 24 h. See SI for full experimental details and conditions.

exemplified by the enantio-retention experiment utilizing 4-Br-*L*-phenylalanine as model substrate (**3a**, Fig. 3A). In contrast, only decomposition product were observed under classic Buchwald-Hartwig C-N coupling conditions (Fig. S2). In addition, elimination-prone coupling partners bearing alkyl chloride (**3b**) were suitable coupling partners. The alkyl halides would valuable for subsequent synthetic applications. Furthermore, heteroaryl rings (Fig. 3B), such as oxazole (**3c**), benzothiazole (**3d**), indazole (**3e**) and imidazole (metronidazole derivative

**3f**), which would be decomposed under strong basic conditions[66], could be well tolerated in our system. This result inspired us to pursue the three-step synthesis of PET (the positron emission computed tomography) tracer analog **3g** for Alzheimer's disease (AD) in 66% overall yield.

Different $B_2N_4$ reagents[67,68] were next used to explore the applicability of these reagents in forming diversified amino products (Fig. 4). $B_2N_4$ reagents derived from pyrrolidine and piperidine could be utilized to construct corresponding *N,N*-dialkylanilines (**4a-4d**). For example, electron-rich and electron-deficient aryl bromides were well tolerated when reacting with tetrakis(pyrrolidino)diborane under standard conditions (**4a-4c**). Furthermore, more valuable tertiary aniline products (**4e-4g**) were successfully prepared with good to excellent yields using corresponding $B_2N_4$ reagents derived from several secondary amines. Unfortunately, aliphatic primary amine-derived diboron species didn't work under standard conditions, most likely due to the undesired hydrolysis of diboron species. For example, only benzyl amine was detected when utilizing $B_2(BnNH)_4$ as $B_2N_4$ reagent. To our delight, pharmaceutical-related diarylamines could be prepared from $B_2(NHPh)_4$ with (hetero)aryl bromides (**4h-4j**). Of note is that more sterically hindered $B_2N_4$ reagents were also good partners to afford diarylamine product (**4k**).

The broad functional group tolerance for this dimethylamination of haloarenes encouraged us to test this strategy for late-stage functionalization of complex molecules, such as natural products and active pharmaceutical ingredients (APIs). As shown in Fig. 5, *N,N*-dimethyl anilines derived from gallic acid (**5a**), thymol (**5c-5d**), estradiol (**5e**), tyramine (**5i**), as well as a variety of APIs (**5b, 5f-5h, 5j-5l**) bearing polar groups, basic heteroatoms, and heterocyclic frameworks were readily synthesized, confirming versatility of our method. Furthermore, a wide variety of powerful synthetic transformations of the resulting *N,N*-dimethyl complexes were also demonstrated using *N,N*-dimethyl aniline derivatives (Fig. S5). These valuable transformations revealed the possibility of global derivatization around the core *N,N*-dimethylaniline structure, and such derivatization can turn *N,N*-dimethylaniline into a versatile synthetic building block in organic synthesis.

## Mechanistic studies

The mechanism of this Ni-catalyzed C-N coupling of haloarenes with $B_2N_4$ type reagents was then investigated. A series of probing experiments were conducted to investigate the reaction pathway. No desired products were observed with other potential intermediates such as dimethylamine, *N,N*-dimethylformamide and tris(dimethylamino)borane (Fig. S7). In the radical trapping experiments (Fig. 6A), when phenyl *tert*-butyl nitrone (PBN) was introduced into the experiments as a spin-trapping reagent, radical EPR signals were observed (Fig. 6Aa). A g-factor of 2.005 indicates that an organic radical species, whose adduct with PBN was detected by high-resolution mass spectrometry (HRMS [M + K]$^+$: 513.3418), was generated from mixture of $Ni(acac)_2$ and $B_2N_4$ reagent after heating. In fact, radical intermediate A1 was identified by oxygen free high-resolution mass spectrometry (HRMS [M + H]$^+$: 298.2716), confirming the single electron transfer progress between Ni(II) and $B_2N_4$ reagent. Notably, the C-N coupling reaction was completely suppressed using TEMPO as a radical inhibitor, and the adduct of TEMPO-acac was detected by HRMS (Fig. 6Ab; [M + H]$^+$: 256.1914). We speculated that TEMPO abstracted acac radical[69] from key radical cage Ni(acac), releasing non-catalytic nickel(0). Additionally, no obvious inhibition of C-N bond formation was observed using 1,1-diphenylethylene as a neutral radical scavenger, indicating that no radical species was involved in the Ni redox cycle (Fig. 6Ac). In this trapping experiment, a key radical intermediate was captured by 1,1-diphenylethylene (HRMS [M + H]$^+$: 479.3736).

Next, we performed several experiments to confirm whether the dominant productive pathway involves a Ni(0)/Ni(II) or Ni(I)/Ni(III) redox cycle. As shown in Fig. 6B, when utilizing $Ni(COD)_2$ as the Ni(0) catalyst, instead of C-N coupling product **2a**, homo-coupling product **2a′** (room temperature or 80 °C) from 4-bromophenoxybenzene was isolated as the single product through the well-established M(0)/M(II) redox cycle (i.e., Negishi coupling[70], Suzuki-Miyaura coupling[51,52]), revealing that Ni(0) was not involved in the Ni redox cycle. Proceeded at room temperature, no desired C-N coupling product **2a** was detected no matter Ni(0) or Ni(II) was used. Well-defined examples of comproportionation reactions[71] of Ni(0) and Ni(II) to generate Ni(I) species have been reported since Heimbach's pioneering work[72]. Based on these general comproportionation strategies of Ni(0) and Ni(II), after 4 h reaction of Ni(0) and Ni(II) ($Ni(COD)_2$: $Ni(acac)_2$ = 20 mol%: 20 mol%) at room temperature, the Ni mixture was added into the reaction system, the desired C-N coupling product was detected with 5% yield (80% yield at 80 °C), indicating that an active Ni(I) species was formed through comproportionation reaction between Ni(0) and Ni(II). The formation of the Ni(I) species was evidenced by electron paramagnetic resonance (EPR) spectroscopy, which revealed a Ni(I) species with $g_{iso}$ = 2.366 from the comproportionation reaction (Fig. 6C, C1). We also conducted EPR measurements on the reaction mixture obtained under standard reaction conditions after 10 h. The unpaired electron in Ni(I) was directly observed by EPR ($g_{iso}$ = 2.349 at 100 K Fig. 6C, C2). While the EPR of C1 is clean, that of C2 is likely affected somewhat by the presence of other organic radicals, for example, A1. Perhaps more straightforwardly, radical intermediate A1 and related fragments A2-A4 (m/z of [M+H]$^+$: 298.2716 for A1; m/z of [M+Na]$^+$: 256.0756 for A2; m/z of [M+H]$^+$: 199.1612 for A3; m/z of [M+H]$^+$: 101.1249 for A4; m/z of [M+Na]$^+$: 290.9448 for A5) and key Ni(III) intermediates (m/z of [M-acac]$^+$: 305.9188 for B; m/z of [M+H]$^+$: 371.1028 for C) were also identified by oxygen free high-resolution mass spectrometry (see supporting information for detailed oxygen free HRMS experiments) after 1 h under standard reaction conditions (Fig. 6D), which supports the potential energy surface obtained from the following density functional theory (DFT) calculations, indicating a Ni(I)/Ni(III) redox cycle was involved in the nickel catalyzed C-N coupling of haloarene.

Computational studies were performed to gain an in-depth understanding of the reaction mechanism (Fig. 7, Supplementary Computational Details). $Ni(acac)_2$, $B_2(NMe_2)_4$, and PhBr were selected as model substrates for this theoretical study. Based on the results of mass spectrometry experiments, we conducted DFT calculations to evaluate the feasibility of forming the $B_2(NMe_2)_4(acac)$ radical species A1 and Ni$^I$(acac) via single electron transfer (SET) process between $Ni(acac)_2$ and $B_2(NMe_2)_4$. Additionally, we also carried out a comprehensive assessment of the solvents' role in our reaction (Fig. S26). The DFT results indicate that the Gibbs free energy (ΔG) for this reaction is 6.4 kcal/mol, suggesting that the reaction is thermodynamically feasible. After the SET process, DMF binds to Ni$^I$(acac) to form intermediate **Int1**, and then oxidative addition occurs via transition state **TS1**, resulting in the Ni(III) intermediate **Int2** with an activation barrier of 13.1 kcal/mol. Next, the Ni(III) intermediate **Int2** coordinates with $B_2(NMe_2)_4$ to form **Int3**. Due to boron's metalloid property, **Int3** undergoes a transmetallation-like four-membered ring transition state **TS2** (25.1 kcal/mol) to form **Int4**. In fact, intermediate **C** (m/z of [M + H]$^+$: 371.1028 in Fig. 6D), derivative of **Int4**, has been detected by HRMS. **Int4** then produces the Ni(I) intermediate **Int5** via the reductive elimination transition state **TS3** (20.3 kcal/mol). Finally, another molecule of DMF replaces $PhNMe_2$ in **Int5** to yield the aniline product and regenerate **Int1** for the next catalytic cycle.

Based on these findings and previous reports[40-47,73-80], we propose the following mechanism (Fig. 8). First, single electron transfer progress between $Ni(acac)_2$ and $B_2(NMe_2)_4$ (**A**) generates $B_2(NMe_2)_4$ radical species **A1** and Ni$^I$(acac). Next, Ni$^I$(acac) undergoes oxidative addition with the aryl bromide **1a**, leading to the formation of Ni(III) species **B**. Subsequent amination of **B** with $B_2(NMe_2)_4$ delivers **C** and

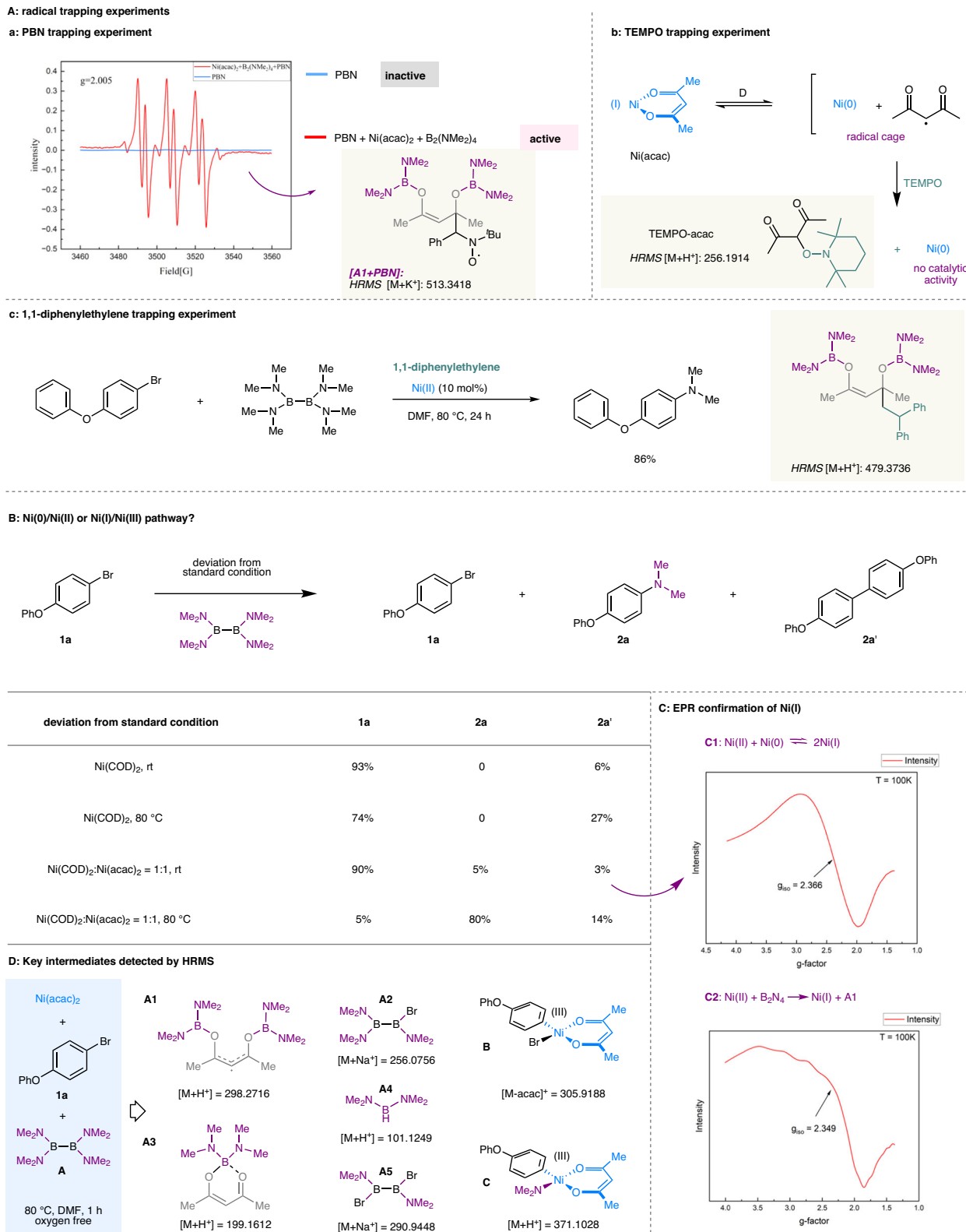

**Fig. 6 | Mechanistic studies of nickel catalyzed C-N coupling of haloarene with B₂(NMe₂)₄.** **A** Radical trapping experiments: (Aa) PBN trapping experiment; (Ab) TEMPO trapping experiment; (Ac) 1,1-diphenylethylene trapping experiment. PBN phenyl *tert*-butyl nitrone; TEMPO 2,2,6,6-Tetramethyl-1-piperidinyloxy; EPR electron paramagnetic resonance. **B** Ni(I)/Ni(III) pathway confirmation. **C** Ni(I)

confirmation through comproportionation reaction (C1) and standard reaction condition (C2) by electron paramagnetic resonance (EPR) experiments in DMF solvent. **D** Key intermediates detected by HRMS. See SI for full experimental details and conditions.

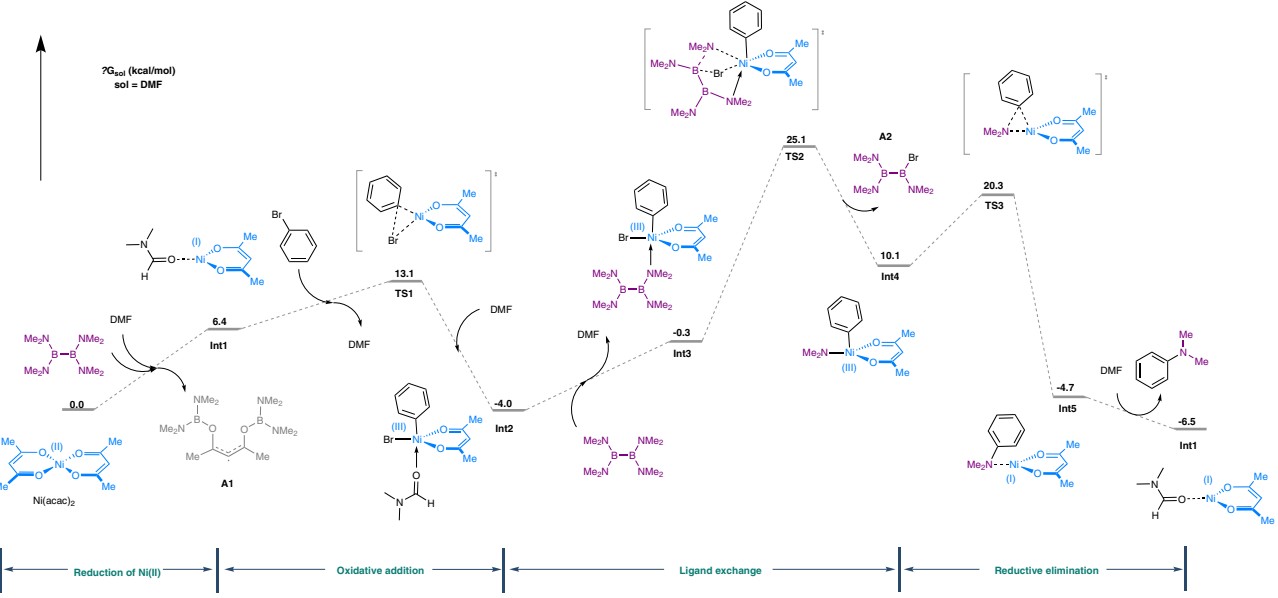

**Fig. 7 | Energy profile of reduction of Ni(II) species and subsequent Ni(I)/Ni(III) catalytic cycle.** A1: $B_2(NMe_2)_4(acac)$ radical species; A2: $B_2(NMe_2)_3Br$; int intermediate, TS transition state.

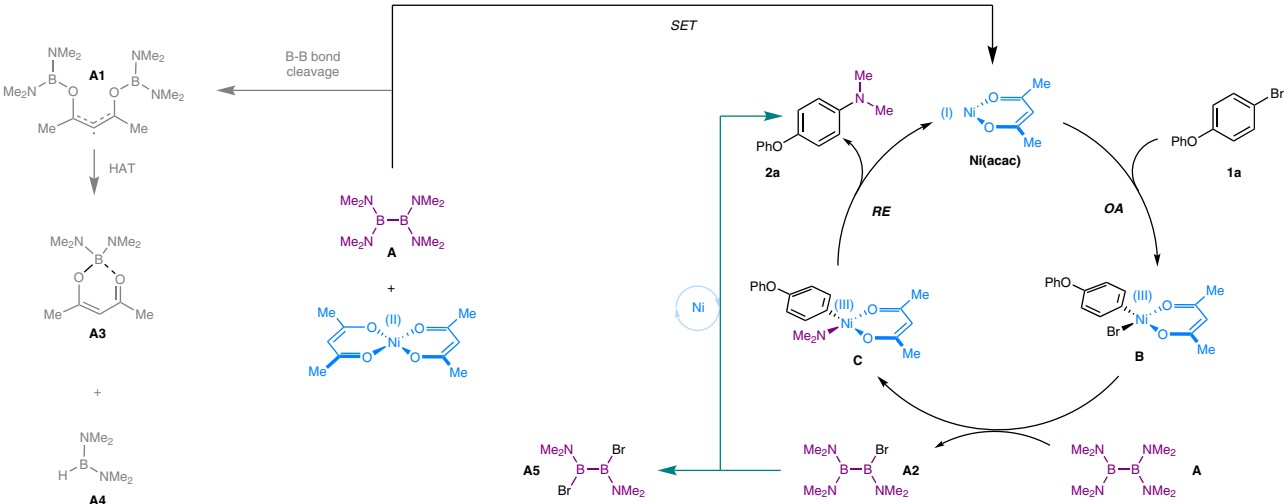

**Fig. 8 | Proposed reaction mechanism of nickel catalyzed C-N coupling of haloarene with $B_2(NMe_2)_4$.** A: $B_2(NMe_2)_4$; A1: $B_2(NMe_2)_4(acac)$ radical species; A2: $B_2(NMe_2)_3Br$; A3: $B(NMe_2)_2(acac)$; A4: $HB(NMe_2)_2$; A5: $B_2(NMe_2)_2Br_2$; HAT hydrogen atom transfer, OA oxidative addition, RE reductive elimination.

**A2**. Feasible reductive elimination of **C** enables the formation of product **2a** and regenerates Ni(I) catalyst.

Additionally, **A2** is also responsible for **2a** synthesis through Ni(I)/Ni(III) redox cycle (green line). 85% yield of **2a** can be isolated with 0.7 equivalent of $B_2(NMe_2)_4$, indicating that, in addition to $B_2(NMe_2)_4$, other dimethylamine sources such as **A2** participate in the C-N bond formation reaction. The by-product (**A5**) of **A2** has been captured by HRMS, showcasing the fact that **A2** is one of the dimethylamine sources.

In conclusion, we have developed an efficient C-N cross coupling methodology for *N,N*-dialkylation of haloarenes with $B_2N_4$ reagents by complex ligand-free and base-free nickel catalysis without an external reductant. Compared with the traditional *N,N*-dialkylaniline synthetic strategies, this protocol benefits from high functional group tolerances, reduced synthetic steps and feasible base sensitive aniline derivatives syntheses. The protocol's excellent functional group tolerance enables the functionalization of a variety of biologically

relevant compounds, representing a valuable potential industrial application of the simple nickel catalyst system. A detailed mechanistic investigation and DFT calculations give a reasonable explanation for the Ni(I)/Ni(III) redox pathway.

## Methods

### General procedure for nickel catalyzed C-N coupling of haloarenes with $B_2N_4$ reagents

An oven-dried 4 mL vial was charged with a magnetic stir bar, aryl bromides (0.2 mmol, 1.0 equiv.), Ni(acac)$_2$ (0.02 mmol, 10 mol%), $B_2N_4$ (0.2 mmol, 1.0 equiv.), DMF (0.5 mL) in the glove box. The vial was sealed with a plastic cap and then stirring was achieved by placing the assembled reactor at 80 °C on IKA C-MAG HS 7 control magnetic stir bars for 24 h. After reaction completion, the reaction was quenched with $H_2O$ and diluted with EtOAc. The resulting mixture was separated and extracted with EtOAc (three times). The combined organic layer was dried over anhydrous $Na_2SO_4$, filtered and concentrated *in vacuo*.

The reaction mixture was purified by fresh silica gel chromatography to afford the desired product.

## Data availability

The experimental and analytical procedures and full spectral data are available in the Supplementary Information. Cartesian coordinates of the calculated structures are available from Supplementary Data 1. Data supporting the findings of this manuscript are also available from the corresponding author upon request.

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

## Acknowledgements

We gratefully acknowledge the financial supports from the Shenzhen Key Laboratory of Neural Cell Reprogramming and Drug Research (No. ZDSYS20230626091202006 to L.W.), the National Natural Science

Foundation of China (No. 22271316 to L.W.; No. 21933004 to Y.-D.W.; No. 22403067 to T.-Y.S.), the Guangdong Basic and Applied Basic Research Foundation (No. 2022A1515011994 to L.W.) and the Fundamental Research Funds for the Central Universities (No. 23ykbj010 to L.W.). Computational studies were supported by Shenzhen Bay Laboratory Supercomputing Center. We also thank Dr. He Zhiqi for helpful discussion.

## Author contributions

Q.C. and L.W. discovered the reactions. Q.C. and Q.L. performed the reaction optimizations, studied the reaction scope and synthetic utility and studied the reaction mechanisms. Y.-H.D. and T.-Y.S. performed the computational study. L.W. wrote the manuscript with help from T.-Y.S. and Y.-D.W. L.W., T.-Y.S., and Y.-D.W. initiated the project, designed the experiments, and directed the research.

## Competing interests

The authors declare no competing interests.
