## [Transparent Peer Review file · Nature Communications]

Nickel catalyzed C-N coupling of haloarenes with B₂N₄ reagents

Corresponding Author: Professor Leifeng Wang

Version 0:

Reviewer comments:

Reviewer #1

(Remarks to the Author)

Wang and Wu have developed an efficient thermal nickel-catalyzed C-N cross-coupling strategy utilizing tetrakis(dimethylamino)diboron and aryl halides as coupling partners under base-free conditions. This methodology demonstrates a broad scope of aryl halides with excellent functional group tolerance. Notably, the method can also be applied for late-stage functionalization and modification of bioactive molecules, highlighting the potential application of this catalytic system. The authors conducted various control experiments, EPR experiments, and DFT calculations to propose a Ni(I)/Ni(III) reaction mechanism. However, despite the technological advancements achieved, this reviewer finds the reported results insufficient for the aims and standards of Nature Communications.

Additional Comments:

1. The authors demonstrated a wide range of aryl halide scope; however, only tetrakis(dimethylamino)diboron was used as the nucleophilic coupling partner. What about other amine-derived diboron species, such as those derived from pyrrolidine, piperidine, benzylamine, etc.? Additionally, is this methodology applicable to primary amine-derived diboron species?
2. TEMPO was used as a radical inhibitor to probe the formation of radical intermediates. However, it is now widely recognized that TEMPO is not an ideal radical inhibitor. The authors should consider including radical clock experiments, 1,1-diphenylethylene as a neutral radical scavenger, and PBN as a radical trap in their EPR experiments, which are all effective for probing radical intermediates. Furthermore, did the authors detect or isolate any TEMPO-coupled adducts?
3. According to the proposed reaction mechanism, the acac ligand plays a crucial role in Ni(I) formation through the SET process. However, when other nickel catalysts such as NiCl₂·6H₂O, Ni(phen)Cl₂, NiBr₂(phen)₂, and Ni(OTf)₂ were used, the desired cross-coupling products were still obtained in moderate to good yields. Could the authors provide a reasonable rationale for this observation?
4. DMF is typically a basic solvent and is known to coordinate with boron, activating it to form boronyl radical species under thermal conditions. Did the authors consider this activation effect in their DFT calculations and mechanistic proposal?
5. The manuscript initially discusses N-methyl anilines, but the study focuses on the C-N cross-coupling of aryl halides with tetrakis(dimethylamino)diboron to produce N,N-dimethyl anilines. This could mislead readers into thinking about N-monomethyl anilines. Additionally, references 2-3, 14 are not relevant to the topic. Please verify all references cited in the manuscript.
6. The manuscript requires careful polishing and reorganization, as some words and sentences are not used correctly. Please review the entire manuscript and supporting information (SI) for issues such as "prefunctionalization," "dominant strategies," and "is dominant."
7. What do the authors mean by "cleaner N,N-dimethyl anilines synthesis"? In terms of atom economy, this is not a cleaner synthesis, as stoichiometric amounts of byproducts are formed, and only 1/4 of the N,N-dimethyl units are incorporated into the target product.
8. In the optimization conditions, the description of entry 10 is followed directly by entry 4, which is not ideal.
9. Table S2 lacks optimization results for transition-metal screening, and Table S4 does not include results for condition screening with NiBr₂(DME).
10. "eq" should be corrected to "equiv."; a space should be added between "80" and "°C." Please check the entire manuscript and SI, as the "°C" symbol is not used correctly.
11. IR spectral data should be included for novel compounds such as 2z, 2ad, 2ap, 3a, 3b, 4b-4e, 4g-4q, 5a. Additionally, melting point data should be provided for novel solid compounds.
12. The NMR spectra for compounds 2m and 2am show some impurities.

Reviewer #2

(Remarks to the Author)

The authors devised and developed an efficient Ni(I)/Ni(III)-catalyzed C-N bond formation of bromoarenes and chloroarenes with B₂(NMe₂)₄ to yield aniline products with good to excellent yields and nice functional group tolerance under mild reaction conditions. This reaction does not require the involvement of bases can promote to efficiently react with electron rich haloarenes and base sensitive substrates. No complex ancillary ligands are required, only simple acac and dme ligands are needed to complete the reaction of bromoarenes and chloroarenes, respectively. The equivalent B₂(NMe₂)₄ diboron reagent can be served as both SET reagent and nitrogen resource in the nickel catalyzed C-N coupling. The Ni(I)/Ni(III) redox cycle could be triggered by the facile SET between Ni(II) and B₂(NMe₂)₄. Based on control experiments, as well as HRMS and EPR detection, several possible intermediates are proposed to involve into the reaction. DFT calculations evaluated the proposed the Ni(I)/Ni(III) redox cycle. The Ni(I)/Ni(III) cycle composed of classic oxidative addition and reductive elimination sounds like reasonable. As such, I am supportive of publication in Nature Communications after some following revisions.

- The SET process is the key to triggering the Ni(I)/Ni(III) cycle, but the so-called "facile" description lacks quantitative or qualitative description except for thermodynamic evaluation. The diboron compound is a common and useful SET reagent in organic photoredox catalysis. How does the B₂(NMe₂)₄ reagent trigger the SET that can reduce Ni(II)? The authors should clearly give the criteria for whether the corresponding SET can occur. Is a thermodynamic exothermic SET process sufficient? Is there any experimental support for electrochemistry, or is thermodynamics sufficient?

- The ligand exchange in the Ni(I)/Ni(III) cycle is full of challenges. The energy barrier exceeds 30 kcal/mol. Considering that the yield is significantly diminished by the reduced loading of B₂(NMe₂)₄, can the diboron intermediates (A and Int4) generated in situ during the reaction participate in the reaction to reduce the energy barrier?

Minor point: Figure 7 and Figure S7 are duplicated.

Reviewer #3

(Remarks to the Author)

The authors developed a nickel-catalyzed C-N bond formation reaction between aryl halides and B₂(NMe₂)₄ to yield aniline products with good to excellent yields under base-free and mild reaction conditions. Additionally, B₂(NMe₂)₄ is utilized as both a single-electron transfer (SET) reagent and a dimethylamine source.

There have been several reports on the construction of C(sp²)-N bonds, among which the methods enabled by nickel catalysis or light-promoted nickel catalysis are particularly attractive (Nat. Catal. 2024, 7, 733; Angew. Chem. Int. Ed. 2021, 60, 21536; Angew. Chem. Int. Ed. 2021, 60, 16077). The method presented in this manuscript provides another pathway to access aniline products. From another perspective, the fact that B₂(NMe₂)₄ is used as both a SET reagent and a dimethylamine source represents an interesting approach. The value of mechanism exploration extends beyond the synthetic value.

Therefore, this reviewer believes that the publication of this manuscript could be considered after the suitable extension of the substrate scopes and the exploration of synthetic applications.

Other comments:

- 1) According to the mechanism mentioned in the text, it is not explicitly stated whether B₂(NMe₂)₄, after losing one NMe₂, can continue to act as a dimethylamine source and provide additional NMe₂.
- 2) The types of B₂N₄ diboron reagents used are too limited.
- 3) The article demonstrates examples involving aryl bromides and aryl chlorides. It is unclear whether this method can be applied to alkenyl bromides or alkynyl bromides.

Reviewer #4

(Remarks to the Author)

In this manuscript, Wang and coworkers described an efficient C-N coupling methodology for N, N-dimethylation of haloarenes with B₂(NMe₂)₄ with a quite broad substrate scope and excellent functional group compatibility under concise nickel catalysis. It is worth noting that this base free and complex ligand free single nickel catalytic system could address the scope limitations when using substrates that would readily undergo undesired racemization, elimination and decomposition reactions in the presence of strong bases. Logical and clear mechanistic investigation and DFT calculation indicate the facile C-N coupling method is operated through a Ni(I)/Ni(III) redox pathway. The N-type Suzuki cross coupling strategy in this manuscript can serve as a nice complement for the efficient clean synthesis of structurally complex anilines to some extent. From the viewpoint of conceptual novelty, in my eyes, is high. Considering both the synthetic utility and the novelty of the method, it is an important contribution in amine synthesis. The test and analysis procedures are detailed and reliable, conclusion is clear, and needs minor revisions before acceptance.

Comment 1:

Seminal strategies for C-N bond construction such as Buchwald's work or others should not be concluded, and related references should be cited (JACS 2024, <https://doi.org/10.1021/jacs.4c09667>; JACS 2024, <https://doi.org/10.1021/jacs.4c08130>).

Comment 2:

In condition screen, is green solvent such as water suitable for this method?

Comment 3:

In the scope, include substrates bearing iodide and bromo groups such as 1-bromo-4-iodidebenzene to demonstrate selectivity? Is naked NH₂ tolerated under standard condition?

Comment 4:

Is this base free C-N cross coupling strategy effective for alkyl C-N construction? For example, using bulky 1-phenethyl bromide as substrate.

Comment 5:

The total yield of 3g should be signed in Figure 4.

Comment 6:

In the computational section, Int1, Int2, ... and TS1, TS2 should be in bold.

Comment 7:

Which intermediate detected by HRMS corresponds to Int4? Please provide a clearer explanation.

Comment 8:

ESI: 1) Table S2, it seems the screen of solvent is meaningless, though the result is obvious. It should be removed from ESI.
2) page 146, chemical structure of 5d and ¹H-NMR spectrum didn't match.

Version 1:

Reviewer comments:

Reviewer #1

(Remarks to the Author)

As the authors have made significant revisions to the previous manuscript, addressing the majority of the comments provided. As a result, I am pleased to recommend the revised version for publication in Nature Communications. However, a few minor issues still need to be resolved before final acceptance:

1. The authors conducted EPR experiments to probe the radical species formed during the reaction, but they described the observed species in general terms (e.g., organic radicals and active) rather than providing the exact chemical formulas and what species was actually captured. Did the authors capture the acac radical, or was it another species? Additionally, in the EPR spectra, the blue color used to represent PBN appears as light blue in some instances. These discrepancies should be corrected for consistency.

2. The EPR spectra of C2 in Figure 7 appear quite different from those of C1, even though the same Ni(I) signal was analyzed. This discrepancy must be carefully investigated, as it represents the most critical point from my perspective. Furthermore, based on previous reports (e.g., ACS Catalysis, 2024, 14, 6897–6914; Nature Catalysis, 2023, 6, 244–253), the EPR signals of Ni(I) typically feature a sharp peak with additional smaller peaks. Could the observed signal instead represent a boronyl radical, acac, or another species? Including a simulated signal in the analysis would provide further clarity.

3. There are several typographical errors in the figures: in Figures 1a and 1b, "transitional metal" should be corrected to "transition metal," and "reductent" should be revised to "reductant." Similarly, in Figure 7C, "confirmaton" should be corrected to "confirmation."

4. The authors noted no reaction occurred under standard conditions with aliphatic primary amine-derived diboron species. A brief explanation of the potential reasons for this observation would be a valuable addition to the manuscript.

5. For all compounds containing fluorine atoms, the coupling constants in the carbon NMR spectra should be explicitly resolved and reported.

Once these issues are addressed, the manuscript will be ready for publication."

Reviewer #2

(Remarks to the Author)

I am very satisfied with the authors' revision and response. My concerns have been considered correctly. I enjoyed the manuscript, as well as the comments for the other three referees. Hence, I would like to recommend this work to be published in Nature Communications.

Minor points for SI:

- The title of Scheme S1 is not appropriate.
- The titles of Scheme S2 and Figure S22 need to be formatted consistently in the same case.
- The title of Figure S19 is missing a period at the end.

Reviewer #3

(Remarks to the Author)

The authors have made extensive revisions and additions to the original manuscript, particularly achieving significant enhancements in the understanding of the mechanisms and notably expanding the substrate scope. On this basis, this reviewer recommends the current version of revised manuscript for publication in Nature Communications.

Reviewer #4

(Remarks to the Author)

Thank you for the revised version. The authors have clarified all the previous concerns. A direct publication in Nature Communication is recommended.

Version 2:

Reviewer comments:

Reviewer #1

(Remarks to the Author)

I am pleased with the authors' revision and response. My concerns have been considered correctly. Hence, I would like to recommend this work to be published in Nature Communications.

Responses to the Comments by Referee: 1

General Comments:

- Wang and Wu have developed an efficient thermal nickel-catalyzed C-N cross-coupling strategy utilizing tetrakis(dimethylamino)diboron and aryl halides as coupling partners under base-free conditions. This methodology demonstrates a broad scope of aryl halides with excellent functional group tolerance. Notably, the method can also be applied for late-stage functionalization and modification of bioactive molecules, highlighting the potential application of this catalytic system. The authors conducted various control experiments, EPR experiments, and DFT calculations to propose a Ni(I)/Ni(III) reaction mechanism. However, despite the technological advancements achieved, this reviewer finds the reported results insufficient for the aims and standards of Nature Communications.

Our Response:

We really appreciate your valuable and constructive comments on our manuscript. We have seriously considered each of your comments, and have carried out additional experiments and calculations, based on which we have carefully revised our manuscript. Some key points are summarized as follows:

(1) We have studied more B₂N₄ reagents, so that the range of aromatic C-N bonds has been expanded considerably. More specifically, in addition to *N,N*-dimethyl, other secondary amines and aromatic primary amines can also be applied to the construction of aromatic C-N bonds. These results are given in Figure 5. Based on these explorations, the title has been changed to "Nickel catalyzed C-N coupling of haloarenes with B₂N₄ reagents".

(2) We have carried out additional experiments to capture free radical intermediates. Several key intermediates were captured, which help the modification of the reaction mechanism.

(3) Additional calculations have been carried out to study the reaction mechanism. A modified potential energy surface has been obtained, as given in Figure 8, which interprets the experimental observations better.

(4) The reaction mechanism has been modified based on the above-mentioned new experimental observations and the calculations, as shown in Figure 9.

We feel that the revised manuscript has been improved significantly. A point-to-point responses to your comments are given below. We hope that these responses are satisfactory. Of course, we would be happy to address any additional concerns from you about the manuscript.

Special Comment 1:

-The authors demonstrated a wide range of aryl halide scope; however, only tetrakis(dimethylamino)diboron was used as the nucleophilic coupling partner. What about other amine-derived diboron species, such as those derived from pyrrolidine, piperidine, benzylamine, etc.? Additionally, is this methodology applicable to primary amine-derived diboron species?

Response 1:

Thank you for your constructive comment. According to your suggestion, we have explored the scope of C-N bond formation using several typical B₂N₄ reagents. Secondary amines such as pyrrolidine, piperidine, and other secondary amines. These reagents work well with the reaction, and give good to excellent yields. Aromatic primary amines such as phenylamine and 2,6-dimethyl phenylamine also work reasonably well, giving moderate yields. These results are given in an added Figure 5. Unfortunately, aliphatic primary amines, such as benzylamine, do not work well. Overall, this additional study broadened the scope of B₂N₄ reagents considerably.

Figure 5. C-N bond formations with diverse B₂N₄ reagents.

Special Comment 2:

-TEMPO was used as a radical inhibitor to probe the formation of radical intermediates. However, it is now widely recognized that TEMPO is not an ideal radical inhibitor. The authors should consider including radical clock experiments, 1,1-diphenylethylene as a neutral radical scavenger, and PBN as a radical trap in their EPR experiments, which are all effective for probing radical intermediates. Furthermore, did the authors detect or isolate any TEMPO-coupled adducts?

Response 2:

Thank you so much for pointing out this critical issue. According to your comments, we have carried out additional experiments to capture free radical intermediates. As shown in revised Figure 7A (given below), when phenyl *tert*-butyl nitron (PBN) was mixed with Ni(acac)₂ and B₂(NMe₂)₄, an EPR signal (g-

factor: 2.005) was observed (Figure **7Aa**) after heating. In addition, a radical intermediate **A1** was identified by oxygen free high-resolution mass spectrometry (HRMS [M+H⁺]: 298.2716). These suggest that the initial step of single electron transfer progress can be between Ni(acac)₂ and B₂(NMe₂)₄. Next, the C-N coupling reaction was completely suppressed using TEMPO as a radical inhibitor, and TEMPO-acac adduct was detected by HRMS (HRMS [M+H⁺]: 256.1914) (Figure **7Ab**). We speculated that TEMPO abstracted acac radical from key radical cage Ni(acac) (ref 69: *Prog. Org. Coat.* **2021**, *151*, 105926), resulting in a non-catalytic nickel(0). (*Ni(0) was unable to catalyze the reaction at reaction condition screen stage*). Furthermore, when we used 1,1-diphenylethylene as a neutral radical scavenger, the C-N coupling reaction was not affected (Figure **7Ac**), indicating that no radical species was involved in the Ni(I)/Ni(III) redox cycle. As a matter of fact, in this trapping experiment, radical intermediate **A1** was captured by 1,1-diphenylethylene (HRMS [M+H⁺]: 479.3736). These three radical trapping experiments, along with the detection of several other radical intermediates and calculations (Figure **8**) allow us to propose a reaction mechanism that is shown in Figure **9**.

Figure 7A. Radical trapping experiments.

Figure 8. Revised energy profile of reduction of Ni(II) species and subsequent Ni(I)/Ni(III)

Figure 9. Revised reaction mechanism of nickel catalyzed C-N coupling of haloarene with $B_2(NMe_2)_4$.

Special Comment 3:

-According to the proposed reaction mechanism, the acac ligand plays a crucial role in Ni(I) formation through the SET process. However, when other nickel catalysts such as $NiCl_2 \cdot 6H_2O$, $Ni(phen)Cl_2$, $NiBr_2(phen)_2$, and $Ni(OTf)_2$ were used, the desired cross-coupling products were still obtained in moderate to good yields. Could the authors provide a reasonable rationale for this observation?

Response 3:

Thank you for your insightful comment. The acac ligand plays a crucial role in the formation of Ni(I) through the single electron transfer (SET) process, facilitating the generation of an intermediate **Int1**. Notably, other nickel catalysts without the acac ligand can also undergo reactions. To gain further insights, we conducted additional calculations employing $Ni(phen)Cl_2$ and $Ni(OTf)_2$ as model catalysts, focusing on the difference in the SET process, and DMF solvent was also considered in stabilizing the boronyl radical through coordination (*J. Am. Chem. Soc.* **2018**, *140*, 6221-6225.) (Figure **S22**). Our computational results revealed that the energy required for SET in $Ni(acac)_2$ is 6.4 kcal/mol (Figure **8**), for $Ni(phen)Cl_2$, the reaction is endothermic by 16.7 kcal/mol, while for $Ni(OTf)_2$, it is endothermic by 16.2 kcal/mol. These do suggest that the latter two reactions are accessible at the reaction temperature, but are less readily than using $Ni(acac)_2$. This information is given in Supporting information (Table **S1**). An intermediate, designated as **A6** in Figure **S21**, was identified using oxygen-free high-resolution mass spectrometry (HRMS $[M+K^+]$: 173.0413), aligning with the initial stage of the SET process. This suggests that $B_2(NMe_2)_4$, upon losing an electron, may form intermediate **A6** and a boronyl radical, which could be potentially stabilized by DMF coordination. Therefore, when other nickel catalysts were used, the desired cross-coupling products were still obtained in moderate to good yields.

Figure S22. Energy Profiles of SET Processes for Ni(phen)Cl₂ and Ni(OTf)₂.

Special Comment 4:

-DMF is typically a basic solvent and is known to coordinate with boron, activating it to form boronyl radical species under thermal conditions. Did the authors consider this activation effect in their DFT calculations and mechanistic proposal?

Response 4:

We have conducted a thorough reassessment of the solvents' role in our reaction, with a particular focus on their influence on the stabilization of boronyl radical species and nickel coordination. After conducting further calculations for Ni(acac)₂ catalyst, we examined the interaction between DMF and the boronyl radical, ultimately determining that no further energy reduction could be obtained through this interaction, as an additional 25.9 kcal/mol is required for DMF to coordinate with the boronyl radical in **A1** (Figure **S19 (3)**). Nevertheless, for the SET process of Ni(phen)Cl₂, the stabilizing effect of DMF on the boronyl radical significantly reduces the reaction energy from 26.0 kcal/mol to 16.7 kcal/mol (Figure **S19 (4)&(5)**). It is possible that the acac ligand confers greater stability to the boronyl species, as the conjugated structures within the acac ligand can potentially enhance the stability of radical species **A1**. Our findings also revealed that the coordination of the solvent to nickel does enhance the stability of the nickel intermediates. Specifically, the ΔG of the intermediate **Int1** is 6.4 kcal/mol, while the energy ΔG of the intermediate **Int1'** is 8.1 kcal/mol (Figure **S19 (1)&(2)**). Furthermore, the coordination of DMF can also stabilize the Ni(III) intermediate **Int2** (Figure **8**). Taking into account your constructive comments, the mechanism of this C-N strategy has become more reasonable and clearer overall.

Figure S19. Reaction energies of reduction of Ni(II) species and plausible stabilizing effect of DMF and acac ligand

Special Comment 5:

-The manuscript initially discusses N-methyl anilines, but the study focuses on the C-N cross-coupling of aryl halides with tetrakis(dimethylamino)diboron to produce N,N-dimethyl anilines. This could mislead readers into thinking about N-monomethyl anilines. Additionally, references 2-3, 14 are not relevant to the topic. Please verify all references cited in the manuscript.

Response 5:

We are grateful for this helpful advice. We feel sorry for the improper citing. Based on your comments, we have verified all cited references and substituted the unrelated references with ref 2 and ref 14. Furthermore, we added some references: 1) seminal strategies for C-N bond construction (ref 25, 27, 29 and 45); 2) B₂N₄ preparation methods (ref 67 and 68); 3) tempo trapping experiments of acac radical (ref 69).

Special Comment 6:

- The manuscript requires careful polishing and reorganization, as some words and sentences are not used correctly. Please review the entire manuscript and supporting information (SI) for issues such as “prefunctionalization,” “dominant strategies,” and “is dominant.”

Response 6:

Thank you for your kind suggestion. We are sorry for this kind of mistakes. Based on your suggestion, we have carefully reviewed the entire manuscript and supporting information (SI) and made the corrections. Additionally, the manuscript has been carefully polished and reorganized.

Special Comment 7:

- *What do the authors mean by “cleaner N,N-dimethyl anilines synthesis”? In terms of atom economy, this is not a cleaner synthesis, as stoichiometric amounts of byproducts are formed, and only 1/4 of the N,N-dimethyl units are incorporated into the target product.*

Response 7:

Thank you for your nice consideration. We originally thought that our method was green for C-N bond formation with base free and mild conditions. However, as you said, its atom economy is not high enough to be called “cleaner”. We have deleted “cleaner” in the manuscript.

Special Comment 8:

- *In the optimization conditions, the description of entry 10 is followed directly by entry 4, which is not ideal.*

Response 8:

Thank you so much for your careful check. Based on your comment, we have reorganized the description of entry **10**.

Special Comment 9:

- *Table S2 lacks optimization results for transition-metal screening, and Table S4 does not include results for condition screening with NiBr₂(DME).*

Response 9:

Thanks for your careful checks. This is really a nice suggestion. According to your kind suggestion, transition-metal screening and condition screening with NiBr₂(DME) results have been added in Table **S1**.

Special Comment 10:

- *“eq” should be corrected to “equiv.”; a space should be added between “80” and “°C.” Please check the entire manuscript and SI, as the “°C” symbol is not used correctly.*

Response 10:

Great thanks for helping us improving our manuscript. We have made all the corrections in our manuscript and SI based on your kind suggestions.

Special Comment 11:

- *IR spectral data should be included for novel compounds such as 2z, 2ad, 2ap, 3a, 3b, 4b-4e, 4g-4q, 5a. Additionally, melting point data should be provided for novel solid compounds.*

Response 11:

We are grateful for this helpful suggestion. According to your advice, we have supplemented the IR spectral data of all novel compounds and melting point data of novel solid compounds.

Special Comment 12:

- *The NMR spectra for compounds 2m and 2am show some impurities.*

Response 12:

Thank you so much for your careful check. Based on your comments, we have substituted the poor spectra.

Responses to the Comments by Referee: 2

General Comments:

- The authors devised and developed an efficient Ni(I)/Ni(III)-catalyzed C-N bond formation of bromoarenes and chloroarenes with $B_2(NMe_2)_4$ to yield aniline products with good to excellent yields and nice functional group tolerance under mild reaction conditions. This reaction does not require the involvement of bases can promote to efficiently react with electron rich haloarenes and base sensitive substrates. No complex ancillary ligands are required, only simple acac and dme ligands are needed to complete the reaction of bromoarenes and chloroarenes, respectively. The equivalent $B_2(NMe_2)_4$ diboron reagent can be served as both SET reagent and nitrogen resource in the nickel catalyzed C-N coupling. The Ni(I)/Ni(III) redox cycle could be triggered by the facile SET between Ni(II) and $B_2(NMe_2)_4$. Based on control experiments, as well as HRMS and EPR detection, several possible intermediates are proposed to involve into the reaction. DFT calculations evaluated the proposed the Ni(I)/Ni(III) redox cycle. The Ni(I)/Ni(III) cycle composed of classic oxidative addition and reductive elimination sounds like reasonable. As such, I am supportive of publication in Nature Communications after some following revisions.

Our Response:

We really appreciate your valuable and constructive comments on our manuscript. We have seriously considered each of your and other reviewers' comments, and have carried out additional experiments and calculations, based on which we have carefully revised out manuscript. Some key points are summarized as follows:

(1) We have studied more B_2N_4 reagents, so that the range of aromatic C-N bonds has been expanded considerably. More specifically, in addition to *N,N*-dimethyl, other secondary amines and aromatic primary amines can also be applied to the construction of aromatic C-N bonds. These results are given in Figure 5. Based on these explorations, the title has been changed to "Nickel catalyzed C-N coupling of haloarenes with B_2N_4 reagents".

(2) We have carried out additional experiments to capture free radical intermediates. Several key intermediates were captured, which help the modification of the reaction mechanism.

(3) Additional calculations have been carried out to study the reaction mechanism. A modified potential energy surface has been obtained, as given in Figure 8, which interprets the experimental observations better.

(4) The reaction mechanism has been modified based on the above-mentioned new experimental observations and the calculations, as shown in Figure 9.

We feel that the revised manuscript has been improved significantly. A point-to-point responses to your comments are given below.

Special Comment 1:

- The SET process is the key to triggering the Ni(I)/Ni(III) cycle, but the so-called "facile" description lacks quantitative or qualitative description except for thermodynamic evaluation. The diboron compound is a common and useful SET reagent in organic photoredox catalysis. How does the B₂(NMe₂)₄ reagent trigger the SET that can reduce Ni(II)? The authors should clearly give the criteria for whether the corresponding SET can occur. Is a thermodynamic exothermic SET process sufficient? Is there any experimental support for electrochemistry, or is thermodynamics sufficient?

Response 1:

We thank the reviewer for pointing out this critical issue. We extend our apologies for the misuse of the term 'facile' in our previous description. The SET process is indeed pivotal in initiating the Ni(I)/Ni(III) catalytic cycle. Given that the reaction is conducted at 80 °C and the SET process is not significantly endothermic based on our DFT calculations ($\Delta G=6.4$ kcal/mol, Figure 8), we consider this process to be viable. We have supplemented the experiments to make a clearer understanding of the initial SET process. Firstly, phenyl *tert*-butyl nitron (PBN) was introduced into the experiments as a spin-trapping reagent (Figure 7Aa), a g-factor in 2.005 indicates that an organic radical species was generated from mixture of Ni(acac)₂ and B₂N₄ reagent after heating. Additionally, radical intermediate **A1** was identified by oxygen free high-resolution mass spectrometry (HRMS [M+H⁺]: 298.2716), matching with the initial step of SET process. Next, SET process was confirmed by radical trapping experiments. The reaction was completely suppressed using TEMPO as a radical inhibitor, and the adduct of TEMPO-acac was detected by HRMS (Figure 7Ab, HRMS [M+H⁺]: 256.1914). We speculated that TEMPO abstracted acac radical from key radical cage Ni(acac) (ref 69: *Prog. Org. Coat.* **2021**, 151, 105926), releasing non-catalytic nickel(0). Additionally, radical intermediate **A1** was captured by 1,1-diphenylethylene (HRMS [M+H⁺]: 479.3736). Above all, the SET process of this C-N strategy is confirmed based on your constructive comments. We have also carried out calculations on the effect of solvent on the stabilities of various intermediates in our proposed mechanism. We found that solvent DMF can coordinate with Ni(III) intermediate, **Int2** (Figure 8), but is replaced by B₂(NMe₂)₄ in **Int3**. A new reaction potential surface has been obtained, which is given in Figure 8.

Figure 8. Revised energy profile of reduction of Ni(II) species and subsequent Ni(I)/Ni(III) catalytic cycle.

Figure 7A. Radical trapping experiments.

Special Comment 2:

- The ligand exchange in the Ni(I)/Ni(III) cycle is full of challenges. The energy barrier exceeds 30 kcal/mol. Considering that the yield is significantly diminished by the reduced loading of $B_2(NMe_2)_4$, can the diboron intermediates (A and Int4) generated in situ during the reaction participate in the reaction to reduce the energy barrier?

Response 2:

A: dominant pathway

B: possible reaction pathway:

Figure S16. Dimethylamine source confirmation

Figure S20. Energy profile of ligand exchange process considering A1 and A2.

Thank you very much for pointing out this issue. (1) We have carried out additional experiments with different diboron loading. As shown in Figure **S16**, no desired product was detected with 0.25 equivalents of $B_2(NMe_2)_4$, suggesting that a portion of $B_2(NMe_2)_4$ is consumed during the initial SET process. The yield increased to 42% with 0.5 equivalents of $B_2(NMe_2)_4$ and significantly improved to 85% with 0.7 equivalents. Notably, yields of 88% were achieved with 0.8 equivalent of $B_2(NMe_2)_4$, indicating that, in addition to $B_2(NMe_2)_4$, other dimethylamine sources indeed participate in the C-N bond formation reaction. (2) We were able to capture a by-product (**A5**: HRMS $[M+Na^+]$: 290.9448) of

A2, showcasing the fact that **A2** is one of the dimethylamine sources (Figure **S16B**). (3) We calculated the likelihood of **A1** and **A2** participating in the ligand exchange process. As shown in Figure **S20A**, DFT calculations indicate that **A2**'s involvement in the reaction reduces the energy of the transmetallation-like four-membered ring transition state from 25.1 kcal/mol to 21.0 kcal/mol. Conversely, **A1**'s participation slightly increases the energy barrier of the rate-determining step, from 25.1 kcal/mol to 25.5 kcal/mol (Figure **S20B**). Combining the above, it is highly plausible that **A2** acts as an additional dimethylamine source in the reaction. However, considering the low concentration of **A2** in the reaction system, we have still used $B_2(NMe_2)_4$ as the dimethylamine source in the rate-determining step of the energy profile, but mentioned “Additionally, **A2** is also responsible for **2a** synthesis through Ni(I)/Ni(III) redox cycle (gray line). 85% yield of **2a** can be isolated with 0.7 equivalent of $B_2(NMe_2)_4$, indicating that, in addition to $B_2(NMe_2)_4$, other dimethylamine sources such as **A2** participate in the C-N bond formation reaction. The by product (**A5**) of **A2** has been captured by HRMS, showcasing the fact that **A2** is one of the dimethylamine sources.” in the last paragraph of the main text.

Special Comment 3:

- *Figure 7 and Figure S7 are duplicated.*

Response 3:

Thank you so much for your careful check. I am very sorry for this kind of mistake. According to your suggestion, we have corrected it.

Responses to the Comments by Referee: 3

General Comments:

- The authors developed a nickel-catalyzed C-N bond formation reaction between aryl halides and $B_2(NMe_2)_4$ to yield aniline products with good to excellent yields under base-free and mild reaction conditions. Additionally, $B_2(NMe_2)_4$ is utilized as both a single-electron transfer (SET) reagent and a dimethylamine source.

There have been several reports on the construction of $C(sp^2)$ -N bonds, among which the methods enabled by nickel catalysis or light-promoted nickel catalysis are particularly attractive (*Nat. Catal.* 2024, 7, 733; *Angew. Chem. Int. Ed.* 2021, 60, 21536; *Angew. Chem. Int. Ed.* 2021, 60, 16077). The method presented in this manuscript provides another pathway to access aniline products. From another perspective, the fact that $B_2(NMe_2)_4$ is used as both a SET reagent and a dimethylamine source represents an interesting approach. The value of mechanism exploration extends beyond the synthetic value.

Therefore, this reviewer believes that the publication of this manuscript could be considered after the suitable extension of the substrate scopes and the exploration of synthetic applications.

Our Response:

We sincerely appreciate your valuable comments. We thank you so much for providing us relevant key articles in the field of Ni-catalyzed C-N bond formation such as (*Nat. Catal.* 2024, 7, 733; *Angew. Chem. Int. Ed.* 2021, 60, 21536; *Angew. Chem. Int. Ed.* 2021, 60, 16077; *Chem. Sci.* 2023, 14, 4390). We have revised the introduction part of our manuscript based on these outstanding articles, which were quoted as ref 28, 30, 45 and 48. We have also carried out additional experiments and calculations based your and other reviewers' comments. Some key points are given below:

(1) We have studied more B_2N_4 reagents, so that the range of aromatic C-N bonds has been expanded considerably. More specifically, in addition to *N,N*-dimethyl, other secondary amines and aromatic primary amines can also be applied to the construction of aromatic C-N bonds. These results are given in Figure 5. Based on these explorations, the title has been changed to "Nickel catalyzed C-N coupling of haloarenes with B_2N_4 reagents".

(2) We have carried out additional experiments to capture free radical intermediates. Several key intermediates were captured, which help the modification of the reaction mechanism.

(3) Additional calculations have been carried out to study the reaction mechanism. A modified potential energy surface has been obtained, as given in Figure 8, which interprets the experimental observations better.

(4) The reaction mechanism has been modified based on the above-mentioned new experimental observations and the calculations, as shown in Figure 9.

We feel that the revised manuscript has been improved significantly. A point-to-point responses to your comments are given below.

Special Comment 1:

- According to the mechanism mentioned in the text, it is not explicitly stated whether $B_2(NMe_2)_4$, after losing one NMe_2 , can continue to act as a dimethylamine source and provide additional NMe_2 .

Response 1:

Thank you very much for pointing out this issue. (1) We have carried out additional experiments with different diboron loading. As shown in Figure S16, no desired product was detected with 0.25 equivalents of $B_2(NMe_2)_4$, suggesting that a portion of $B_2(NMe_2)_4$ is consumed during the initial SET process. The yield increased to 42% with 0.5 equivalents of $B_2(NMe_2)_4$ and significantly improved to 85% with 0.7 equivalents. Notably, yields of 88% were achieved with 0.8 equivalent of $B_2(NMe_2)_4$, indicating that, in addition to $B_2(NMe_2)_4$, other dimethylamine sources indeed participate in the C-N bond formation reaction. (2) We were able to capture a by-product (**A5**: HRMS $[M+Na^+]$: 290.9448) of **A2**, showcasing the fact that **A2** is one of the dimethylamine sources (Figure S16B). (3) We calculated the likelihood of **A1** and **A2** participating in the ligand exchange process. As shown in Figure S20A, DFT calculations indicate that **A2**'s involvement in the reaction reduces the energy of the transmetalation-like four-membered ring transition state from 25.1 kcal/mol to 21.0 kcal/mol. Conversely, **A1**'s participation slightly increases the energy barrier of the rate-determining step, from 25.1 kcal/mol to 25.5 kcal/mol (Figure S20B). Combining the above, it is highly plausible that **A2** acts as an additional dimethylamine source in the reaction. However, considering the low concentration of **A2** in the reaction system, we have still used $B_2(NMe_2)_4$ as the dimethylamine source in the rate-determining step of the energy profile, but mentioned "Additionally, **A2** is also responsible for **2a** synthesis through Ni(I)/Ni(III) redox cycle (gray line). 85% yield of **2a** can be isolated with 0.7 equivalent of $B_2(NMe_2)_4$, indicating that, in addition to $B_2(NMe_2)_4$, other dimethylamine sources such as **A2** participate in the C-N bond formation reaction. The by product (**A5**) of **A2** has been captured by HRMS, showcasing the fact that **A2** is one of the dimethylamine sources." in the last paragraph of the main text.

A: dominant pathway

B: possible reaction pathway:

Figure S16. Dimethylamine source confirmation

Figure S20. Energy profile of ligand exchange process considering A1 and A2.

Figure 9. Revised reaction mechanism of nickel catalyzed C-N coupling of haloarene with $B_2(NMe_2)_4$.

Special Comment 2:

- The types of B_2N_4 diboron reagents used are too limited.

Response 2:

Thank you for your constructive comment. According to your suggestion, we have explored the scope of C-N bond formation using several typical B_2N_4 reagents. Secondary amines such as pyrrolidine, piperidine, and other secondary amines. These reagents work well with the reaction, and give good to excellent yields. Aromatic primary amines such as phenylamine and 2,6-dimethyl phenylamine also work reasonably well, giving moderate yields. These results are given in an added Figure 5. Overall, this additional study broadened the scope of B_2N_4 reagents considerably.

Figure 5. Exploration of C-N bond formation with diverse B_2N_4 reagents.

Special Comment 3:

- The article demonstrates examples involving aryl bromides and aryl chlorides. It is unclear whether this method can be applied to alkenyl bromides or alkynyl bromides.

Response 3:

Thanks for your nice suggestions. Alkenyl bromides or alkynyl bromides were examined under standard conditions based on your suggestion. Unfortunately, no desired product was isolated using either (bromoethenyl)benzene or (bromoethynyl)benzene. In contrast, alkyl bromides are well tolerated through Ni-catalyzed C-N bond formation, delivering the corresponding tertiary amines with good to high yields. We have also included the corresponding characterization data in the Supporting Information (Figure S6).

Figure S6. Exploration of C-N bond formation with alkyl bromides.

Responses to the Comments by Referee: 4

General Comments:

- In this manuscript, Wang and coworkers described an efficient C-N coupling methodology for N,N-dimethylation of haloarenes with B₂(NMe₂)₄ with a quite broad substrate scope and excellent functional group compatibility under concise nickel catalysis. It is worth noting that this base free and complex ligand free single nickel catalytic system could address the scope limitations when using substrates that would readily undergo undesired racemization, elimination and decomposition reactions in the presence of strong bases. Logical and clear mechanistic investigation and DFT calculation indicate the facile C-N coupling method is operated through a Ni(I)/Ni(III) redox pathway. The N-type Suzuki cross coupling strategy in this manuscript can serve as a nice complement for the efficient clean synthesis of structurally complex anilines to some extent. From the viewpoint of conceptual novelty, in my eyes, is high. Considering both the synthetic utility and the novelty of the method, it is an important contribution in amine synthesis. The test and analysis procedures are detailed and reliable, conclusion is clear, and needs minor revisions before acceptance.

Our Response:

We really appreciate your valuable and constructive comments on our manuscript. We have seriously considered each of your and other reviewers' comments, and have carried out additional experiments and calculations, based on which we have carefully revised our manuscript. Some key points are summarized as follows:

(1) We have studied more B₂N₄ reagents, so that the range of aromatic C-N bonds has been expanded considerably. More specifically, in addition to N,N-dimethyl, other secondary amines and aromatic primary amines can also be applied to the construction of aromatic C-N bonds. These results are given in Figure 5. Based on these explorations, the title has been changed to "Nickel catalyzed C-N coupling of haloarenes with B₂N₄ reagents".

(2) We have carried out additional experiments to capture free radical intermediates. Several key intermediates were captured, which help the modification of the reaction mechanism.

(3) Additional calculations have been carried out to study the reaction mechanism. A modified potential energy surface has been obtained, as given in Figure 8, which interprets the experimental observations better.

(4) The reaction mechanism has been modified based on the above-mentioned new experimental observations and the calculations, as shown in Figure 9.

We feel that the revised manuscript has been improved significantly. A point-to-point response to your comments are given below.

Special Comment 1:

-Seminal strategies for C-N bond construction such as Buchwald's work or others should not be concluded, and related references should be cited (JACS 2024, <https://doi.org/10.1021/jacs.4c09667>; JACS2024, <https://doi.org/10.1021/jacs.4c08130>)?

Response 1:

We thank you so much for providing us latest key articles in the field of C-N bond formation such as (*J. Am. Chem. Soc.* **2024**, *146*, 26609-26615; *J. Am. Chem. Soc.* **2024**, *146*, 26936-26946). We revised the introduction part of our manuscript based on these outstanding articles, which were quoted as ref 25 and 45.

Special Comment 2:

- In condition screen, is green solvent such as water suitable for this method?

Response 2:

Thanks for your comment. Water was used as solvent based on your comments. However, no desired product was isolated. We speculate that hydrolysis of $B_2(NMe_2)_2$ would occur in the presence of water.

Special Comment 3:

- In the scope, include substrates bearing iodide and bromo groups such as 1-bromo-4-iodidebenzene to demonstrate selectivity? Is naked NH_2 tolerated under standard condition?

Response 3:

Thanks for your helpful comments. This is really a nice advice. Based on your suggestion, 1-bromo-4-iodidebenzene was utilized to demonstrate the reaction selectivity. Under standard conditions, 4-bromo-*N,N*-dimethylaniline was isolated as the dominant product, indicating that our method could be used in the selective amination of substrates bearing iodide and bromo groups. Additionally, naked NH_2 was tolerated under standard conditions, yielding the desired product with moderate yield. The NH_2 product was added as **2s** in Figure 3.

Figure 1. New substrates exploration of C-N bond formation.

Special Comment 4:

- Is this base free C-N cross coupling strategy effective for alkyl C-N construction? For example, using bulky 1-phenethyl bromide as substrate?

Response 4:

Thanks for your valuable comments. Several alkyl bromides has been examined utilizing our base free C-N cross coupling strategy, delivering the corresponding tertiary alkylamines with good to excellent yields. Of note is that bulky 1-phenethyl bromide derivative is well tolerated. These results will be published separately.

Figure 2. Exploration of C-N bond formation with alkyl bromides.

Special Comment 5:

- The total yield of 3g should be signed in Figure 4.

Response 5:

Thank you so much for your careful check. According to your advice, we have corrected it.

Special Comment 6:

- In the computational section, *Int1*, *Int2*, ... and *TS1*, *TS2* should be in bold.

Response 6:

We are grateful for this helpful check. Based on your kind suggestion, we have made the correction.

Special Comment 7:

- Which intermediate detected by HRMS corresponds to *Int4*? Please provide a clearer explanation.

Response 7:

Thank you for your critical comment. We are sorry for making it unclear. We have revised the description as "Due to boron's metalloid property, **Int3** undergoes a transmetalation-like four-membered ring transition state **TS2** (25.1 kcal/mol) to form **Int4**. In fact, intermediate **C** (m/z of [M+H⁺]: 371.1028), derivative of **Int4**, has been detected by HRMS".

Special Comment 8:

- ESI: 1) Table S2, it seems the screen of solvent is meaningless, though the result is obvious. It should be removed from ESI. 2) page 146, chemical structure of 5d and 1H-NMR spectrum didn't match.

Response 8:

Thank you so much for your careful check. Table **S2** has been deleted based on your helpful comments. Additionally, the mistake of NMR spectrum has been corrected.

Responses to the Comments by Referee: 1

General Comment:

As the authors have made significant revisions to the previous manuscript, addressing the majority of the comments provided. As a result, I am pleased to recommend the revised version for publication in Nature Communications. However, a few minor issues still need to be resolved before final acceptance:

Specific Comment 1

The authors conducted EPR experiments to probe the radical species formed during the reaction, but they described the observed species in general terms (e.g., organic radicals and active) rather than providing the exact chemical formulas and what species was actually captured. Did the authors capture the acac radical, or was it another species? Additionally, in the EPR spectra, the blue color used to represent PBN appears as light blue in some instances. These discrepancies should be corrected for consistency.

Response 1:

We really appreciate your comments and suggestions concerning our manuscript, which are very helpful for us to improve the manuscript. We have carried out additional experiments to capture some key radical intermediates and further confirmed the proposed reaction mechanism.

As shown in **Figure 7Aa**, we have captured the adduct of organic radical species (**A1**) with PBN [**A1-PBN**] through high-resolution mass spectrometry (HRMS [M+K⁺]: 513.3418) in the PBN trapping experiment. The EPR spectrum of [**A1-PBN**] is quite different from the reported EPR spectrum of the adduct of acac radical with PBN (*Polym. Chem.* **2018**, 9, 1371-1378.) as shown in **Figure A**, and therefore, excluding the possibility of acac radical.

Figure A. Reported EPR spectrum of the adduct of acac radical with PBN.

In fact, radical intermediate **A1** has also been identified by oxygen free high-resolution mass spectrometry (HRMS [M+H⁺]: 298.2716), confirming the single electron transfer process between Ni(II) and B₂N₄ reagent. In addition, the blue color used to represent PBN has been corrected as light blue (Figure 7Aa).

Figure 7. Mechanistic studies of nickel catalyzed C-N coupling of haloarene with B₂(NMe₂)₄. (A) Radical trapping experiments. (B) Ni(I)/Ni(III) pathway confirmation. (C) Ni(I) confirmation through comproportionation reaction (C1) and standard reaction condition (C2) by electron paramagnetic resonance (EPR) experiments in DMF solvent. (D) Key intermediates detected by HRMS. See SI for full experimental details and conditions.

Special Comment 2:

- The EPR spectra of C2 in Figure 7 appear quite different from those of C1,

even though the same Ni(I) signal was analyzed. This discrepancy must be carefully investigated, as it represents the most critical point from my perspective. Furthermore, based on previous reports (e.g., *ACS Catalysis*, 2024, 14, 6897-6914; *Nature Catalysis*, 2023, 6, 244-253), the EPR signals of Ni(I) typically feature a sharp peak with additional smaller peaks. Could the observed signal instead represent a boronyl radical, acac, or another species? Including a simulated signal in the analysis would provide further clarity.

Response 2:

First, we corrected an error in the EPR spectrum of C2 in the previous manuscript. It now covers the range of 4.0-1.5. Now the $g_{\text{iso}} = 2.366$ for C1 and $g_{\text{iso}} = 2.349$ for C2 are quite similar.

Indeed, the EPR spectra of C2 and C1 are somewhat different. This is because in C1 only Ni(I) species is present, and the spectrum is clean. However, in C2 under standard reaction conditions, besides Ni(I) species, other radical species, for example, **A1**, also exist and they might affect the spectrum.

The EPR spectra of Ni(I) in DMF and powder EPR spectrum (*Inorg. Chem.* **2011**, 50, 8630-8635) are quite different as shown in Figure B. In our experiment, we carried out the EPR analysis in DMF solvent, which is different from related Ni(I) powder EPR reports (e.g., *ACS Catal.* **2024**, 14, 6897-6914; *Nat. Catal.* **2023**, 6, 244-253). These related articles have been added as ref. 79 and ref. 80, respectively.

Figure B. Reported EPR spectrum of Ni(I) in DMF and powder EPR spectrum.

Special Comment 3:

- There are several typographical errors in the figures: in Figures 1a and 1b, "transitional metal" should be corrected to "transition metal," and "reductent" should be revised to "reductant." Similarly, in Figure 7C, "confirmaton" should be corrected to "confirmation."

Response 3:

Thank you for your kind suggestion. We are sorry for this kind of mistakes. Based on your suggestion, we have corrected the typographical errors.

Special Comment 4:

- *The authors noted no reaction occurred under standard conditions with aliphatic primary amine-derived diboron species. A brief explanation of the potential reasons for this observation would be a valuable addition to the manuscript.*

Response 4:

We are grateful for this helpful advice. Based on your suggestion, a brief explanation of the potential reasons for unreactive primary amine-derived diboron species under standard conditions has been added in the manuscript: “Unfortunately, aliphatic primary amine-derived diboron species didn’t work under standard conditions, most likely due to the undesired hydrolysis of diboron species. For example, only benzyl amine was detected when utilizing $B_2(BnNH)_4$ as B_2N_4 reagent.”

Special Comment 5:

- *For all compounds containing fluorine atoms, the coupling constants in the carbon NMR spectra should be explicitly resolved and reported.*

Response 5:

Thank you for your kind suggestion. Based on your comment, the coupling constants in the carbon NMR spectra of all compounds containing fluorine atoms have been explicitly resolved and reported.

Thank you again for your positive and constructive comments and suggestions on our manuscript. We hope that the revised manuscript is satisfactory for publication in *Nature Communications*.

Responses to the Comments by Referee: 2

General Comments:

- I am very satisfied with the authors' revision and response. My concerns have been considered correctly. I enjoyed the manuscript, as well as the comments for the other three referees. Hence, I would like to recommend this work to be published in Nature Communications.

Our Response:

We really appreciate your valuable and constructive comments to help our manuscript to be published in *Nature Communications*.

Special Comments:

- Minor points for SI:*
- The title of Scheme S1 is not appropriate.*
- The titles of Scheme S2 and Figure S22 need to be formatted consistently in the same case.*
- The title of Figure S19 is missing a period at the end.*

Response:

Thank you so much for your careful check. According to your suggestions, we have corrected these errors.